# Design of a Novel Compact MICS Band PIFA Antenna for Implantable Biotelemetry Applications

**DOI:** 10.3390/s22218182

**Published:** 2022-10-26

**Authors:** Ziyang Huang, Hao Wu, Seedahmed S. Mahmoud, Qiang Fang

**Affiliations:** 1Department of Biomedical Engineering, Shantou University, Shantou 515063, China; 2Suzhou Institute of Biomedical Engineering and Technology, Chinese Academy of Sciences, Suzhou 215163, China

**Keywords:** implantable antenna, MICS band, PIFA antenna, SAR

## Abstract

An implantable stacked planar inverted-F antenna (PIFA) for biotelemetric communication in the 402–405 MHz Medical Implant Communications Service (MICS) frequency band is designed and fabricated. With the proposed PIFA structure, a slot on each radiating patch was embedded, resulting in a size reduction of 0.013 λ and a compact size of 10 × 10 × 1.905 mm${}^{3}$. Both in vitro and in vivo experiments verified the simulation performance with characteristics of −10 dB bandwidth of 29 MHz, radiation efficiency of 0.9%, and a maximum far-field gain of −18.8 dB. We calculated the safety power delivered to the antenna using the specific absorption rate (SAR) limitation standard. Compared to other implantable antennas for biotelemetry, this antenna performs comparably and has a smaller size. This design would further develop implantable medical devices that communicate in the MICS band.

## 1. Introduction

A wireless communication link between implantable medical devices (IMDs) and external equipment is essential to an intelligent implantable medical system. This type of implantable system can serve either therapeutic or diagnostic purposes. The communication link can temporarily or permanently set the operating parameters, update the embedded program, capture the device status, and record the therapy history for medical implants.

The low-frequency inductive coil-based wireless link is the most popularly-used biotelemetric method for implantable medical devices [1,2]. However, the shortcomings of this approach are a low data rate, limited communication range, and sensitivity to coil alignment. Communication quality can be improved if radio frequency (RF) communication is chosen and the corresponding antennas are designed. At present, various frequency bands have been allocated for implantable antennas, such as Medical Implant Communications Service (MICS) band (402–405 MHz), Medical Device Radio (MedRad) band (401–406 MHz), and Industrial, Scientific, and Medical (ISM) band (2.4–2.5 GHz) [3]. Considering its exceptional suitability for propagation through human tissues, the Medical Implant Communications Service (MICS) band (402–405 MHz) is highly recommended for IMDs [4]. Nevertheless, there is also an antenna size miniaturization problem associated with this frequency band. An antenna of λ/2 or λ/4 size is impractical because the wavelength is large in this frequency band, and so are resonant-type antennas (λ = 744 mm at 403 MHz in vacuum). Thus, the miniaturization technique plays a vital role in MICS antenna design.

Planar Inverted F-Antenna (PIFA) is a common miniaturization technique for antennas in which the upper radiating layer and ground layer are connected with a shorting pin, reducing the antenna’s size by half and altering the resonant frequency from λ/2 to λ/4 [5]. In [6], a spiral-shaped printed PIFA was implemented on a circular substrate, which utilized FR4 epoxy as a dielectric material. A dimension of $\pi \times 9^{2}\times$ 0.8 mm${}^{3}$ and a maximum far-field gain of −30.79 dBi have been achieved. However, this antenna has a limited bandwidth, which might lead to low acceptability against frequency shifting in practical situations. Hosain et al. [7] proposed a circular meandered PIFA with a relatively large volume, a radius of 10 mm, thickness of 1.5 mm, and comparably narrow bandwidth of 16 MHz. Furthermore, in [8], the researchers designed a compact dual-band PIFA by embedding an open-ended L-shaped slot on the radiation patch surface. This antenna achieved a competitive performance with a relatively compact size, wide bandwidths, and typical gain levels. In [9], a compact broadband antenna with a radiating layer consisting of two inverted oblong-shaped open rings and C-shaped stripline was proposed, with a wide bandwidth of 52 MHz but a large volume of 479 mm${}^{3}$. Besides the PIFA structure, Kawdungta et al. [10] also designed a microstrip antenna for ISM and MICS bands. This antenna miniaturized the size by connecting the radiating patch and the ground plane with a loading resistor as the shorting pin, ultimately achieving a dimension of 6 × 28 × 1.27 mm${}^{3}$, and antenna gains of −38 dBi and −13 dBi for MICS and ISM bands, respectively. A circularly polarized patch antenna was designed for pacemaker monitoring in [11]. Even though this design has an acceptable bandwidth, practical gain, and a low thickness of 0.635 mm, it has a relatively large planar size of 40×40 mm${}^{2}$. Based on the aforementioned work, miniaturization, along with improving the performance of antennas, including bandwidth, the maximum far-field gain, and other characteristics, is the common pursuit of implantable antenna design.

In this paper, we proposed a new design of a three-layer stacked PIFA with a parasitic patch layer on top for IMDs operating in the MICS band. Spiral rectangular-shaped slots were etched on the two radiating patch layers, serving as current paths for resonant frequency tuning. The merits of the proposed antenna design include its relatively compact size, competitive far-field gain, and satisfactory bandwidth and return loss level. The miniaturization techniques adopted for this antenna design are first elaborated. The second section demonstrates the parametric study and the antenna performance, followed by the optimal simulation results. To ensure antenna safety, the 1-g averaged Specific Absorption Rate (SAR) was simulated, and according to the SAR limitation, the maximum power to be delivered to the antenna was calculated. In the last section, an antenna prototype was fabricated to verify the performance, and both in vitro and in vivo experiments were conducted. This part presents and discusses the measurement results and the comparison with the simulation.

## 2. Antenna Design

Figure 1 shows the geometry of the proposed miniaturized implantable antenna for biomedical communication in the MICS band. The PIFA-based structure with a parasitic top radiating patch layer achieves a total dimension of 10 × 10 × 1.905 mm${}^{3}$. The PIFA structure has been selected due to its size reduction ability and simplicity of adopting different miniaturizing techniques. The proposed antenna inserted a shorting pin between the ground and the middle layer to increase the effective size of the antenna [12]. We introduced two cutting slots on the middle and top layer patches to extend the current path on radiation patches [13,14], thereby tuning the resonant frequency. These slots resulted in spiral rectangular-shaped strips for both layers. It is evident from the lateral view in Figure 1a that the antenna comprises three stacked substrate layers made of Rogers RO3010 material, which has a permittivity of 10.2, reducing the wavelength and, ultimately, dimensions of the antenna effectively [15]. Each substrate has a size of 10 × 10 mm${}^{2}$ and a thickness of 0.635 mm. The top (Figure 1b) and middle (Figure 1c) radiation patch layers were embedded on the top of the middle and bottom substrates, respectively, which served as radiation elements, while the ground plane shown in Figure 1d was printed on the bottom part of the bottom substrate. Furthermore, the superstrate was placed above the top patch layer to separate the patch from the human tissue to prevent infections, short circuits, and direct exposure to RF energy [16].

The current distributions of the top and middle radiating patches at 403 MHz are shown in Figure 2 to illustrate the patch layer operation mechanism. It is apparent from Figure 2a, that strong currents from the feeder point flow counter-clockwise to the top radiating patch, gradually decreasing in amplitude along the path. This indicates that the top layer patch essentially serves as a resonator at 403 MHz. In Figure 2b, currents flow out of the shorting pin towards the feeder point in the left part, while currents flow on the right side are meandered due to the shorting pin, increasing the length of the current path. The co-effects of lengthening the current paths of both layers conduce to lower the resonant frequency, tuning it to be excited at 403 MHz.

## 3. Parametric Study and Simulation Results

This antenna was designed to operate in the muscle layer at a resonant frequency of 403 MHz in the MICS band. In the simulation, a parameter model of the antenna structure was first built, and a three-layer simple tissue model was subsequently setup, consisting of skin, fat, and muscle. Based on [17,18] and the calculation program of the dielectric properties in body tissues at different frequencies in an online resource [19] of the Institute of Applied Physics (IAP), the size and electric characteristics of three tissue layers at 403 MHz can be obtained (as shown in Table 1). Modeling and parameter analysis were performed using an electromagnetic-field simulation software ANSYS High-Frequency Structure Simulator (HFSS).

### 3.1. Position of Shorting Pin

The position of the shorting pin affects the antenna’s resonant frequency and its match with the feeding system. In this model, the location of the pin, indicated as ($x_{s}$, $y_{s}$) in Figure 1c, was moved along the middle layer strip. By keeping $x_{s}$ in a constant value and altering $y_{s}$ within a preset range, Figure 3 reveals the relation between the shorting pin location along the y-axis and the return loss. As can be observed, resonant frequency (m1, m2, m3) increases as $y_{s}$ increases, while the absolute value of return loss decreases accordingly. As a result, the antenna and feeding system are no longer a match.

### 3.2. Length of Top Spiral Line

A longer conductor line can increase the length of the current path on the radiating patches and, equivalently, the effective size of the PIFA, thereby decreasing its resonant frequency. In Figure 1b, *L* represents the length of the top spiral strip end line, and changing *L* results in changing the length of the entire top spiral strip. Based on this premise, the relation between resonant frequency and variable *L* was demonstrated in Figure 4, where the resonant frequency decreases with the increase of *L*, indicating a worse match.

### 3.3. Optimal Simulation Results

Based on the above parametric studies and empirical approaches, it can be deduced that a 403 MHz resonant frequency occurs when parameter $y_{s}$ and L lie within the ranges of −1.5 mm to −2.3 mm and 2 mm to 2.5 mm, respectively. Both ranges were subsequently fed into the parameter sweep analysis to determine the optimal parameter values to attain the minimum return loss at the resonant frequency of 403 MHz. From the simulation results, the optimal parameter values were found at $y_{s}$ = −2.1 mm and *L* = 2.2 mm. From Figure 5, it can be obtained that the return loss at 403 MHz is −42 dB while the −10 dB bandwidth is 26.2 MHz from 390.1 MHz to 416 MHz, which covers the entire MICS band of 402–405 MHz.

The 1 g average SAR value was calculated, with its distribution shown in Figure 6. The maximum SAR was 702 W/kg, distributed near the antenna’s location. The default input power to the antenna for this simulation was 1 W. According to IEEE C95.1–1999 standard, which specifies the limitation of 1 g average SAR value to be 1.6 W/kg [20], the delivered power of this proposed antenna should be constrained within 2.27 mW.

## 4. Measurement Results and Discussion

### 4.1. In Vitro Experiment

Figure 7a shows the appearance and size of the fabricated antenna. To connect with the test instrument, an SMA connector was used with its outside body bonded on the ground layer patch of the antenna and its centre to the feeder. The in vitro experiment was conducted to test the antenna performance in animal tissue samples. The antenna’s return loss was calculated using an Agilent E5242A network analyzer. A semi-rigid coaxial cable was used to connect the network analyzer and the SMA connector at the back of the antenna. Then the Antenna was buried into the tissue sample, i.e., a pork shoulder as a substitute for human muscle. The pork shoulder was finely chopped to ensure close contact between itself and the antenna, avoiding introducing large air bubbles around the surface area. The measurement setup is shown in Figure 7b.

The measurement result is shown in Figure 8a. The return loss curve indicates that the antenna’s actual resonant frequency is 394 MHz with a minimum return loss of −39 dB and the −10 dB bandwidth from 379 MHz to 408 MHz covering the whole MICS band. It is noted that a 10 MHz deviation from 403 MHz was measured. The same problem also appeared in [13]. However, when the antenna was placed at the surface of the pork with the SMA connector hanging in the air without contacting the pork, the resonant frequency was back to 403 MHz, which is shown in Figure 8b.

To further analyze this phenomenon, the position of the designed antenna was adjusted to various embedded depths. Figure 9a demonstrates different embedded depths using a cross-sectional schematic, and the corresponding results were displayed in Figure 9b. It was found that the increasing depth of the embedded antenna implies a larger contact area between the SMA connecter and the pork meat, and a lower resonant frequency of the antenna. Therefore, it was hypothesized that the SMA connector caused this frequency deviation.

A simple model of the SMA connector (shown in Figure 10a) was added to the original antenna design using the provided datasheet [21] to attest to the possible interference introduced by the connector. As expected, the resonant frequency deviated with a new value of 389 MHz (see Figure 10a), which is close to the measured result of 394 MHz. To further investigate the influence caused by the SMA connector, a variable h was set to a value ranging from 1 mm to 5 mm, and the analysis results are given in Figure 11. The resonant frequency declined (m1–m5) with the increase of the length (h). As a result, this investigation verified the influence of the SMA connector on the antenna. It proved the resonant frequency in the original model was accurate, and the design was effective.

The radiation pattern measurement was conducted in a 4 m × 4 m anechoic chamber equipped with a SATIMO SG24 system. The Antenna Under Test (AUT) was set on the top of the mast, which can rotate horizontally along the central axis. An array of 24 probes was distributed around the arch in the vertical plane driven by a control unit for electric scanning. The testing scenario is shown in Figure 12a, and the horizontal far-field radiation pattern at 403 MHz is given in Figure 12b with a maximum gain of −18.8 dB and radiation efficiency of 0.9%. Given the electrically small size of the antenna and in-body power dissipation [22], this low radiation efficiency is expected. It lies in an acceptable range compared to other published results (0.04–1.12%) [23,24,25,26,27].

### 4.2. In Vivo Experiment

The in vivo experiment was approved by the ethics committee of the University of Chinese Academy of Sciences and was conducted in the Animal Experiment Center at Soochow University. Rabbits were selected as the experimental animal due to sufficient thicknesses of their muscle layers for the antenna placement. As shown in Figure 13a–c, the antenna was initially sterilized with alcohol before the surgery. Then, the rabbit was anesthetized with an injection of 2.5% pelltobarbitalum natricum solution into the vein of its ear. The site for implantation was the bicep muscle on the rabbit’s upper leg. The antenna was buried in the muscle with the SMA bonding post outside the muscle to connect with a coaxial line.

Figure 13d shows the measurement setup intuitively. The return loss was measured immediately after the surgery using an Agilent N5242A PNA network analyzer, with the results in Figure 14. The whole procedure took 20 min to complete. It was found that the antenna’s resonant frequency was 390 MHz, with a return loss value of −20 dB, and the −10 dB bandwidth of 30 MHz from 374 MHz to 404 MHz. The bandwidth still covers the MICS band, with a 13 MHz deviation to the targeted resonant frequency due to the SMA connector’s influence.

### 4.3. Comparative Analysis

To further illustrate the superiority of the proposed antenna, a comparative analysis summarizing the characteristics (frequency bands, dimensions, and maximum far-field gain) of previously-published antennas has been given in Table 2. As can be observed, compared to other similar antennas, the proposed antenna has a relatively smaller volume, maintaining a superb maximum far-field gain in the MICS band. The comparative results suggest that the proposed antenna is competitive with existing prototypes for biomedical implantable use. A specific advantage of the proposed antenna is that it is compact compared to other implantable antennas while maintaining other qualities.

## 5. Conclusions

A miniaturized three-layer PIFA antenna operating in the MICS band has been designed for implantable biomedical communication. Miniaturization techniques have obtained a reduced overall volume of 190.5 mm${}^{3}$. The safety power delivered to the antenna was calculated according to the SAR limitation standard. Through in vitro and in vivo experiments, the actual performance of the antenna was investigated and found to be satisfactory. Through a simulation analysis followed by in vitro experiment, the reason for the resonant frequency shift was found to be the interference caused by the SMT connector. Therefore, the resonant frequency shift problem will not exist for future practical applications.

## Figures and Tables

**Figure 1 sensors-22-08182-f001:**
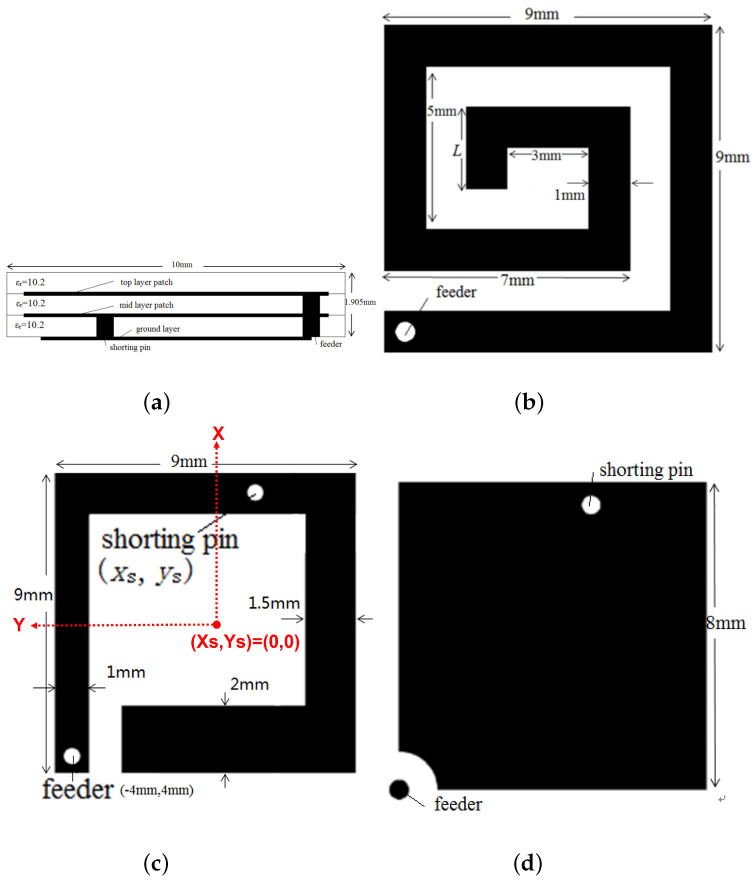
Antenna configuration. (**a**) From the side view, from the top view of (**b**) the top layer, (**c**) the middle layer, and (**d**) the ground layer. All the units are in mm.

**Figure 2 sensors-22-08182-f002:**
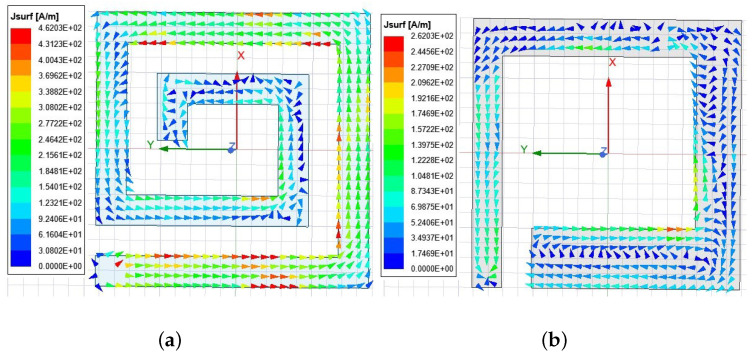
Current distributions of the proposed antenna at 403 MHz. (**a**) Top radiation patch layer; (**b**) middle radiation patch layer.

**Figure 3 sensors-22-08182-f003:**
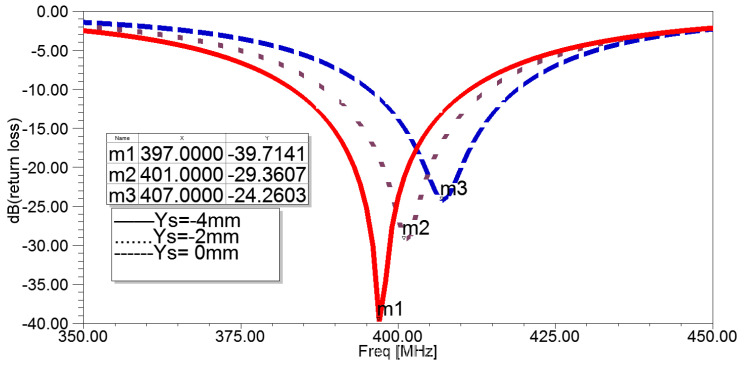
Return losses of the antenna in simulation setups according to different $y_{s}$.

**Figure 4 sensors-22-08182-f004:**
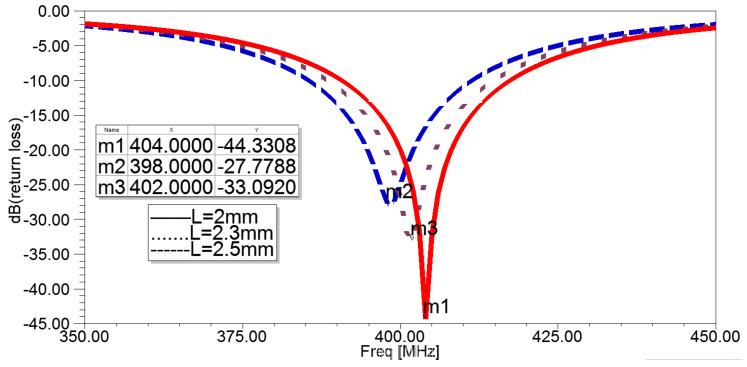
Return losses of the antenna in simulation setups according to different *L*.

**Figure 5 sensors-22-08182-f005:**
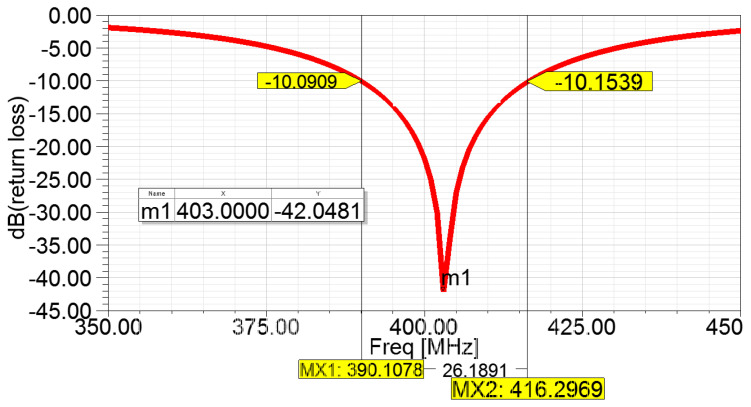
Return loss of final parameter.

**Figure 6 sensors-22-08182-f006:**
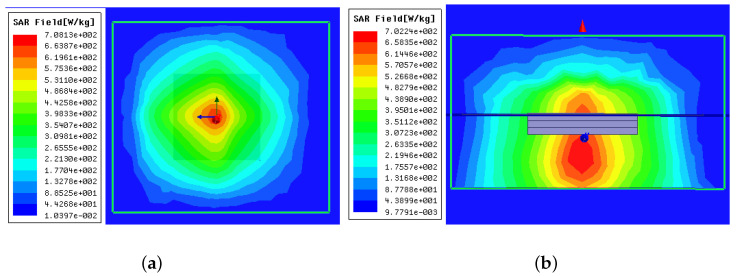
Simulated SAR distribution (**a**) in x-y plane above and the top layer of the antenna and (**b**) x-z plane.

**Figure 7 sensors-22-08182-f007:**
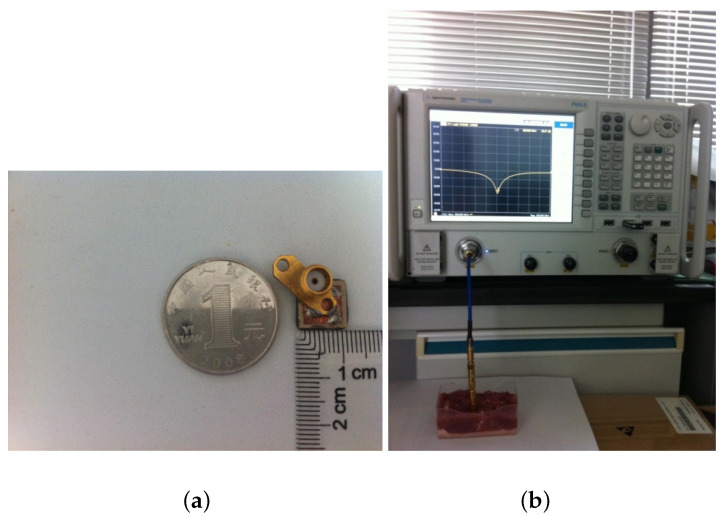
The fabricated prototype and the experimental setup of the proposed antenna. (**a**) The actual fabricated antenna; (**b**) in vitro experiment setup with pork shoulder.

**Figure 8 sensors-22-08182-f008:**
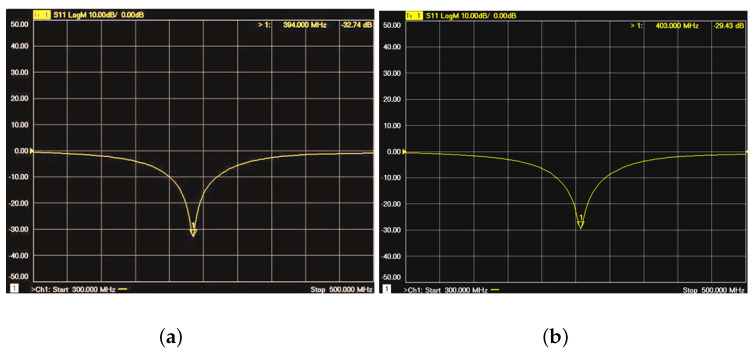
Return losses of the fabricated antenna buried (**a**) in the pork and (**b**) on the surface of pork.

**Figure 9 sensors-22-08182-f009:**
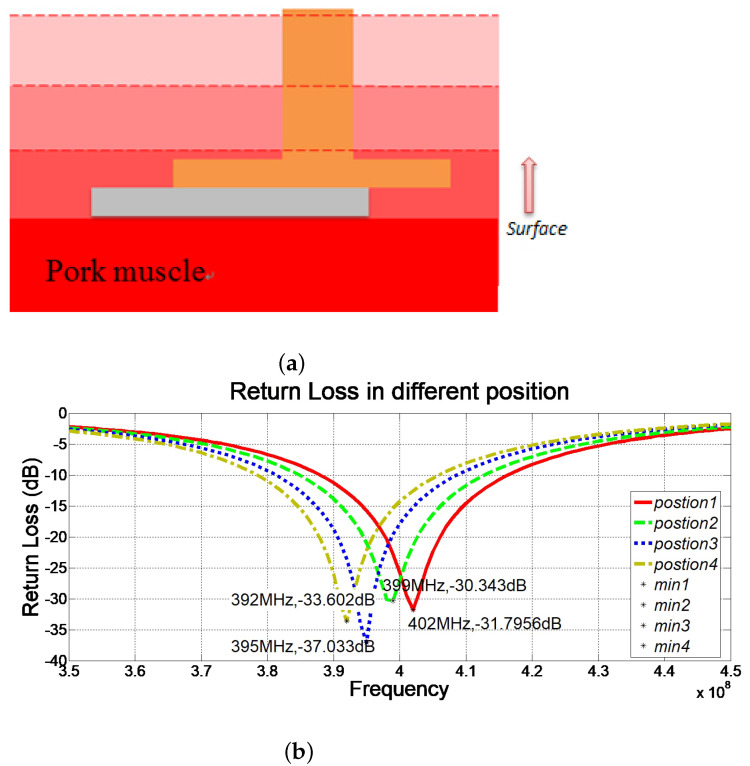
(**a**) Antenna embedded in different depths and (**b**) the corresponding return losses.

**Figure 10 sensors-22-08182-f010:**
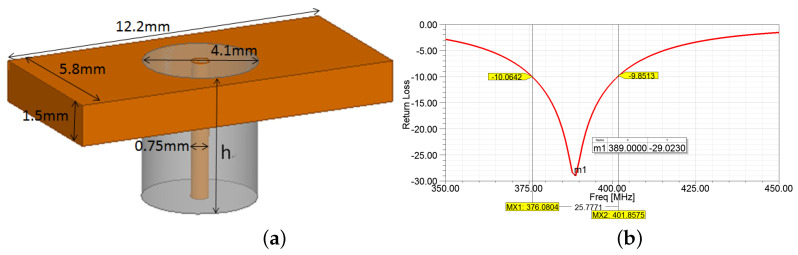
(**a**) A simple model of SMA connector and (**b**) the new return loss.

**Figure 11 sensors-22-08182-f011:**
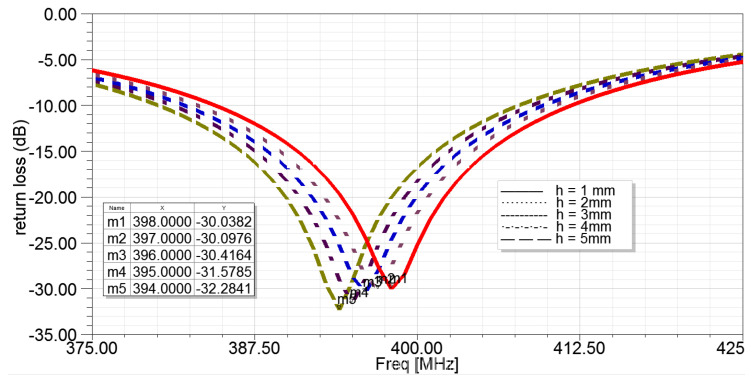
The return losses of the antenna with different values of height (h) of the SMA connector.

**Figure 12 sensors-22-08182-f012:**
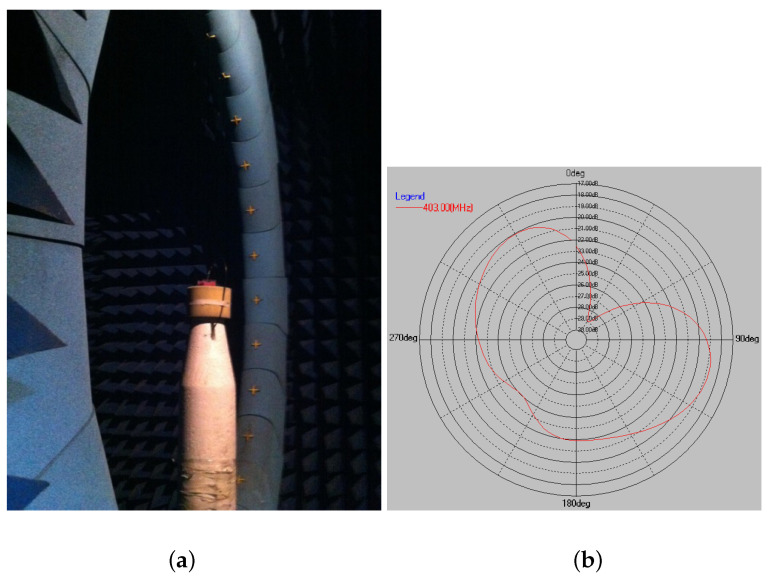
(**a**) The measurement setup inside an anechoic chamber, and (**b**) the horizontal radiation pattern.

**Figure 13 sensors-22-08182-f013:**
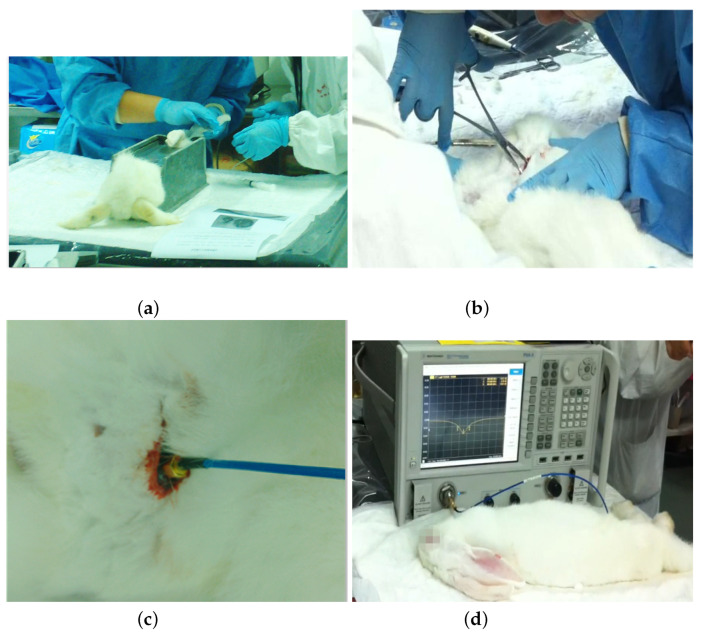
Surgical implantation of the antenna into the rabbit. (**a**) Anesthetizing the rabbit. (**b**) Incision made through the skin. (**c**) Placement of the implantable antenna. (**d**) Measurement setup.

**Figure 14 sensors-22-08182-f014:**
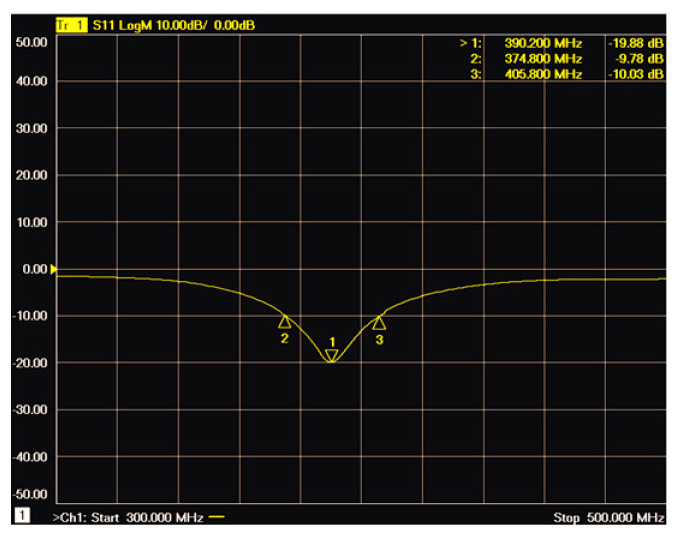
The return loss of the proposed antenna buried in the muscle layer of rabbit.

**Table 1 sensors-22-08182-t001:** Electric properties of human tissue at 403 MHz.

Tissue	Thickness (mm)	Relative Permittivity $\epsilon_{r}$	Conductivity σ (S/m)	Mass Density ρ (kg/m^3^)
Skin	3	46.72	0.68	1100
Fat	10	5.58	0.04	916
Muscle	20	57.10	0.79	1041

**Table 2 sensors-22-08182-t002:** Comparison of characteristics of implantable antennas.

Reference Type	Frequency Band	Dimension (mm${}^{3}$)	Gain (dB)
Palandoken 2017 [28]	MICS/ISM	432 (15 × 15 × 1.92)	MICS: −14.4 ISM: −14.55
Sulaiman 2018 [29]	MICS	4103.1 (30.5 × 21.02 × 6.4)	-
Bakogianni 2019 [8]	MedRadio/ISM	284.48 (16 × 14 × 1.27)	MedRadio: −35.9 ISM: −24.3
Fu 2019 [30]	ISM	242 (11 × 11 × 2)	−22.95
Usluer 2020 [31]	MICS/ISM	248.92 (14 × 14 × 1.27)	MICS: −48.15 ISM: −21.15
Li 2019 [9]	MedRadio	479 (23 × 16.4 × 1.27)	−37.05
Kawdungta 2021 [10]	MICS/ISM	213.36 (6 × 28 × 1.27)	MICS: −40.15 ISM: −15.15
Proposed Antenna	MICS	190.5 (10 × 10 × 1.905)	−18.8

## Data Availability

Not applicable.

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
