# Peer review of "Design of a Novel Compact MICS Band PIFA Antenna for Implantable Biotelemetry Applications"

_sensors, 2022, doi:10.3390/s22218182_

Round 1
Reviewer 1 Report
In this paper, the authors proposed a compact PIFA antenna for biotelemetry application in MICS band 402-405 MHz. The authors fabricated a prototype of the designed antenna and presented some measured results to validate the obtained simulation results. However, the main drawback of the paper is that it has limited novelty, more specifically, the actual research contributions is not cleared to me. My detailed comments are given as follows -
1. The introduction section needs to be updated. With detailed literature survey, the actual contribution with respect to the state-of-the-art relevant published works should be highlighted.
2. The coordinate of the shorting pin position in Fig. 1(c) is not cleared unless the reference coordinate axis. In the figure the reference coordinate axis should be included to understand the reference displacement of the mentioned position. Similarly, without reference coordinate axes it is not possible to understand the actual direction of the variation of the variable ys.
3. The parametric analysis presented in Fig 2 and 3 are corresponding to only with increasing the respective parameter as compared to its optimum value. However, to show the optimum selection of the parameter value, the result for both higher and lower along with the optimum parameter value is desirable.
4. It is mentioned that the substrates is placed above the top layer patch to separate the patch and human tissue to avoid possible infection, short circuit, and direct exposure of RF energy to the human body.
Then, in the designed antenna, the bottom ground plane is unprotected and how such problem is avoided for the ground plane.
5. A comparative table with detailed design dimensions and miniaturization must be included to quantify and illustrate the performance of the proposed antenna with respect to the others.
6. Please check the abbreviation used to define "Institute of Applied Physics".
Author Response
Please see the attached word file, thank you.

Reviewer 2 Report
The manuscript presents an interesting article to lower the resonant frequency. Please note the following:
1. The structure is not exactly PIFA. There is a parasitic element on the top layer. I think the stripline line on the top layer plays an important role but I do not quite understand how it works. I want to see more explanation by showing the simulated field distributions.
2. The radiation efficiency of 0.9% is way too low. Authors are claiming that is more or less a norm, but I am not sure. I like to see some comparison with published results.
Author Response

(The authors gave the same response as above.)

Round 2
Reviewer 1 Report
After addressing all the comments raised by the reviewers, the quality of the manuscript is improved. The manuscript now may be accepted in its present form.
Author Response
Thank you for giving us the opportunity. We have now worked on language problems and have proofread carefully this time. We really hope that the fluency and language level have improved as expected.

Reviewer 2 Report
Now I do see that there are two antenna elements. Those two are strongly coupled. There may be two possible results: widening bandwidth and lowering the resonant frequency. Authors tried to explain the latter but not in a fully convincing manner. For example, let's eliminate the top layer. Does it increase the frequency drastically? If it does, how much. Also how does the modification affect the polarization? Does the polarization really matter in the application?
Author Response
Thank you for giving us additional opportunity to explain more clearly on this specific point. The response can be found in the attached word file, please check it.
